# Rational engineering of minimally immunogenic nucleases for gene therapy

Rumya Raghavan[1,2,3,4,5,6,9], Mirco J. Friedrich[1,2,3,4,5,9], Indigo King[7], Samuel Chau-Duy-Tam Vo[1,2,3,4,5], Daniel Strebinger[1,2,3,4,5], Blake Lash[1,2,3,4,5], Michael Kilian[8], Michael Platten[8], Rhiannon K. Macrae[1,2,3,4,5], Yifan Song[7], Lucas Nivon[7] & Feng Zhang[1,2,3,4,5] ✉

Genome editing using CRISPR-Cas systems is a promising avenue for the treatment of genetic diseases. However, cellular and humoral immunogenicity of genome editing tools, which originate from bacteria, complicates their clinical use. Here we report reduced immunogenicity (Red)(i)-variants of two clinically relevant nucleases, SaCas9 and AsCas12a. Through MHC-associated peptide proteomics (MAPPs) analysis, we identify putative immunogenic epitopes on each nuclease. Using computational modeling, we rationally design these proteins to evade the immune response. SaCas9 and AsCas12a Redi variants are substantially less recognized by adaptive immune components, including reduced binding affinity to MHC molecules and attenuated generation of cytotoxic T cell responses, yet maintain wild-type levels of activity and specificity. In vivo editing of *PCSK9* with SaCas9.Redi.1 is comparable in efficiency to wild-type SaCas9, but significantly reduces undesired immune responses. This demonstrates the utility of this approach in engineering proteins to evade immune detection.

The first CRISPR-based genome editing therapy was recently approved for clinical use, and many more are currently being tested in the clinic to treat a variety of genetic disorders including retinal dystrophy, hemophilia, lysosomal storage disorders, and certain types of cancer[1–7]. These powerful therapies consist of a Cas nuclease and a guide RNA (gRNA) that dictates the target site to be edited[8]. To date, most CRISPR-based therapies rely on one of three Cas nucleases: *Streptococcus pyogenes* Cas9 (SpCas9), *Staphylococcus aureus* Cas9 (SaCas9), and *Acidaminococcus species* Cas12a (AsCas12a)[9–12]. Among these three, SaCas9 and AsCas12a are the focus of most in vivo therapeutic strategies because their size allows them to be packaged into adeno-associated viral (AAV) vectors, the leading delivery modality for in vivo gene therapies[9–11,13].

In addition to efficient delivery of these tools, a second challenge to their in vivo use is their potential immunogenicity, particularly due to their bacterial origin, as many patients have pre-existing exposures and immune responses to microbially-derived molecules[14–16]. It has been reported that 80% of healthy individuals have both humoral immunity, mediated by antibodies, and cellular immunity, mediated by T-cells, against proteins derived from *Staphylococcus aureus* and *Streptococcus pyogenes*[2,17]. This pre-existing immunity extends to Cas nucleases as well: profiling blood from healthy human donors revealed that 78% had class-switched to immunoglobulin (IgG) antibodies against SaCas9 and 58% had antibodies against SpCas9, and all donors that were positive for cellular immunity against Cas9 also had antibody activity, indicating a high concordance between adaptive and humoral immunity[11,16,18–20].

[1]Broad Institute of MIT and Harvard, Cambridge, MA 02142, USA. [2]McGovern Institute for Brain Research at MIT, Cambridge, MA 02139, USA. [3]Department of Brain and Cognitive Science, Massachusetts Institute of Technology, Cambridge, MA 02139, USA. [4]Department of Biological Engineering, Massachusetts Institute of Technology, Cambridge, MA 02139, USA. [5]Howard Hughes Medical Institute, Cambridge, MA 02139, USA. [6]Harvard-MIT Division of Health Sciences and Technology, Massachusetts Institute of Technology, Massachusetts 02139 Cambridge, USA. [7]Cyrus Biotechnology, Seattle, WA 98121, USA. [8]Clinical Cooperation Unit Neuroimmunology and Brain Tumor Immunology, German Cancer Research Center, DKFZ, Heidelberg, Germany. [9]These authors contributed equally: Rumya Raghavan, Mirco J. Friedrich. ✉e-mail: zhang@broadinstitute.org

To address this challenge, we profile SaCas9 and AsCas12a to identify putative immunogenic epitopes and then computationally design variants of these two nucleases predicted to evade immune detection. We experimentally validate these variants, showing that they eliminate CD8+ T cell reactivity in vitro while retaining native levels of nuclease activity and specificity. Additionally, in an immunocompetent MHC class I/II humanized mouse model, we demonstrate that the SaCas9 variants effectively reduce humoral and cellular immune reactions as compared to the wild-type nucleases. These results provide a framework for engineering therapeutic proteins to reduce immunogenicity and provide a starting point for the development of safer CRISPR-based therapeutics.

## Results

### Computational design of non-immunogenic epitopes in SaCas9 and AsCas12a

Although pre-existing immunity to SaCas9 has been reported[16,21,22], the specific epitopes recognized by the immune system are not known. Therefore, we profiled MHC Class I specific peptide sequences mediating CD8+ T cell immunogenicity to both SaCas9 and AsCas12a by performing MHC-associated peptide proteomics (MAPPs) analysis on HLA-A*0201-expressing MDA-MB-231 cells transfected with a plasmid expressing either SaCas9 or AsCas12a[23]. Peptides bound to MHC class I molecules were then identified by mass spectrometry (Fig. 1a and Supplementary Data 1–4). The MAPPs analysis nominated three immunodominant epitopes for each nuclease; for SaCas9, the predicted epitopes were (1) 8-GLDIGITSV-16, (2) 926-VTVKNLDVI-934, and (3) 1034-ILGNLYEVK-1050, and for AsCas12a, the predicted epitopes were (1) 210-RLITAVPSL-218, (2) 277-LNEVLNLAI-285, and (3) 971-YLSQVIHEI-979. Using reported structures of SaCas9 and AsCas12a, we then took a structure-guided computational approach to design mutants that would preserve nuclease activity while ablating or reducing MHC class I binding to a representative set of 14 HLA alleles (Supplementary Table 1). Briefly, we used the Rosetta protein design package to introduce mutations into the nuclease model to reduce MHC-binding propensity of all peptide subsequences around the epitope for a representative allele from every HLA-A, HLA-B, and HLA-C supertype cluster as defined in Rasmussen et al.[24]. This approach aimed to eliminate known MHC-binding epitopes without creating new predicted epitopes while maintaining protein stability. Mutations were evaluated in silico for protein stability and peptide-MHC binding (Fig. 1b). From this process and the immunopeptidomics data analysis, we found that of all the tested MHCs, peptide-MHC binding was most pronounced for HLA-A*0201, and we therefore chose to focus on this allele for in vitro and in vivo work moving forward (Supplementary Figs. 1 and 2). For each immunogenic epitope, we designed three or four variants containing single point mutations (Fig. 1c, d). The locations of the proposed mutations do not overlap with either the DNA or RNA binding regions or the catalytic sites. Furthermore, all mutations were modeled in silico to ensure adequate shape complementarity to avoid clashes, such that no mutations were predicted to disrupt native nuclease function outside the tolerable range (Fig. 1d).

### SaCas9 and AsCas12a peptide variants demonstrate reduced CD8+ T cell reactivity

We used NetMHCpan 4.1, a neural network tool that predicts peptide-MHC class I binding, to verify the findings of the MAPPs analysis and predict whether our proposed point mutant peptides would be presented on MHC class I molecules to CD8+ T cells[25]. Consistent with our MAPPs analysis, the wild-type peptides for SaCas9 epitope 1 as well as AsCas12a epitopes 1 and 3 had a NetMHCpan predicted rank score below 0.5, indicating strong binding (shown as inverted rank score in Fig. 2a). Wild-type peptides for SaCas9 epitopes 2 and 3 as well as AsCas12a epitope 2 were not predicted to be strongly immunogenic by NetMHCpan 4.1. All peptide variants containing single point mutations

were predicted to decrease the binding strength between peptide and MHC. Next, we experimentally assessed the degree of CD8+ T cell mediated immunogenicity in PBMCs derived from healthy human donors to each identified epitope and their peptide variants using an ELISpot assay, which measures T-cell recognition following peptide binding to MHC class I molecules (Fig. 2b). We synthesized peptides of either wild-type or point mutant-containing variants of each epitope and then mixed with peripheral blood mononuclear cells (PBMCs) from healthy donors (HLA-A*0201). For SaCas9, we observed a robust immune response to wild-type peptide epitopes 1 and 2 and a more muted immune response to the epitope 3 peptide. Compared to the wild-type peptide, we observed that the single-mutant peptide variants for epitopes 1 and 2 had produced significantly fewer spots, indicating a reduced immune reaction to these variants. The epitope 3 peptide variants showed comparable levels of spot formation as wild-type epitope 3 (Fig. 2c, d and Supplementary Figs. 3, 4). For AsCas12a, we observed robust spot formation in the presence of wild-type peptide epitopes 1 and 3 but less robust spot formation in the presence of wild-type peptide epitope 2 which was also consistent with our NetMHCpan computational predictions. As with SaCas9, we found that all AsCas12a single-mutant peptide variants significantly reduced spot formation relative to wild-type (Fig. 2c, d and Supplementary Figs. 3, 4). We tested these peptides in PBMCs representative of a wider breadth of HLA-types and found some reduced immunogenicity with the mutant peptides even from HLA backgrounds outside of HLA-A*0201 (Supplementary Figs. 5 and 6). We did not observe any increase in immunogenicity with any of the mutant peptides in any of the non-HLA-A*0201 patients profiled (Supplementary Figs. 5 and 6). Together, these results validate the MAPPs analysis and indicate that the mutant peptide sequences reduce recognition by CD8+ T cells from human donors (who may or may not be naive to SaCas9 or AsCas12a) as compared to the native sequences.

### Evaluation of nuclease efficiency and specificity of SaCas9 and AsCas12a variants

We generated full-length proteins containing the single point mutants and tested their editing efficiency in human cells by transfecting them with nuclease-containing plasmids. For SaCas9, we found that most single mutants exhibited indel activity at or above 60% of the wild-type activity, although certain amino acid substitutions, such as L9F and L1035T, completely ablated nuclease activity (Fig. 3a). To reduce immune reactivity to any of the three immunodominant epitopes identified in SaCas9, we generated double and triple mutants combining the single mutations from across epitopes that maintained high indel activity (see Methods and Supplementary Data 5). We selected the eight of the highest performing triple mutants and profiled their activity across a panel of different target sites. Of these eight mutants, three showed comparable levels of activity to wild-type SaCas9 across the panel of targets: L9A/I934T/L1035A, L9S/I934K/ L1035V and V16A/I934K/L1035V, which we refer to collectively as reduced immunogenicity (Redi) variants and specifically as SaCas9.Redi.1, .2, and .3, respectively (Fig. 3b).

We repeated this process for AsCas12a and again observed that most single point mutants maintained nuclease activity, except L211V, L211A, and I285V (Fig. 3c). After testing double mutants (Supplementary Data 6), we generated triple mutants and profiled their indel activity at a single target (Supplementary Data 6). We expanded our analysis of the top seven triple mutants by testing their indel activity at six target sites. Based on these results, we selected three triple mutants for further characterization: L218S/I285S/L972A, L218S/I285T/L972A, and L218T/I285S/L972A, which we refer to as AsCas12a.Redi.1, .2, and .3, respectively (Fig. 3d). We then used Tagmentation-based Tag integration site sequencing (TTISS) to confirm that the SaCas9.Redi and AsCas12a.Redi variants maintained specificity by assessing their genome-wide off-target effects, finding no significant differences from wild-type (Fig. 3e)[26].

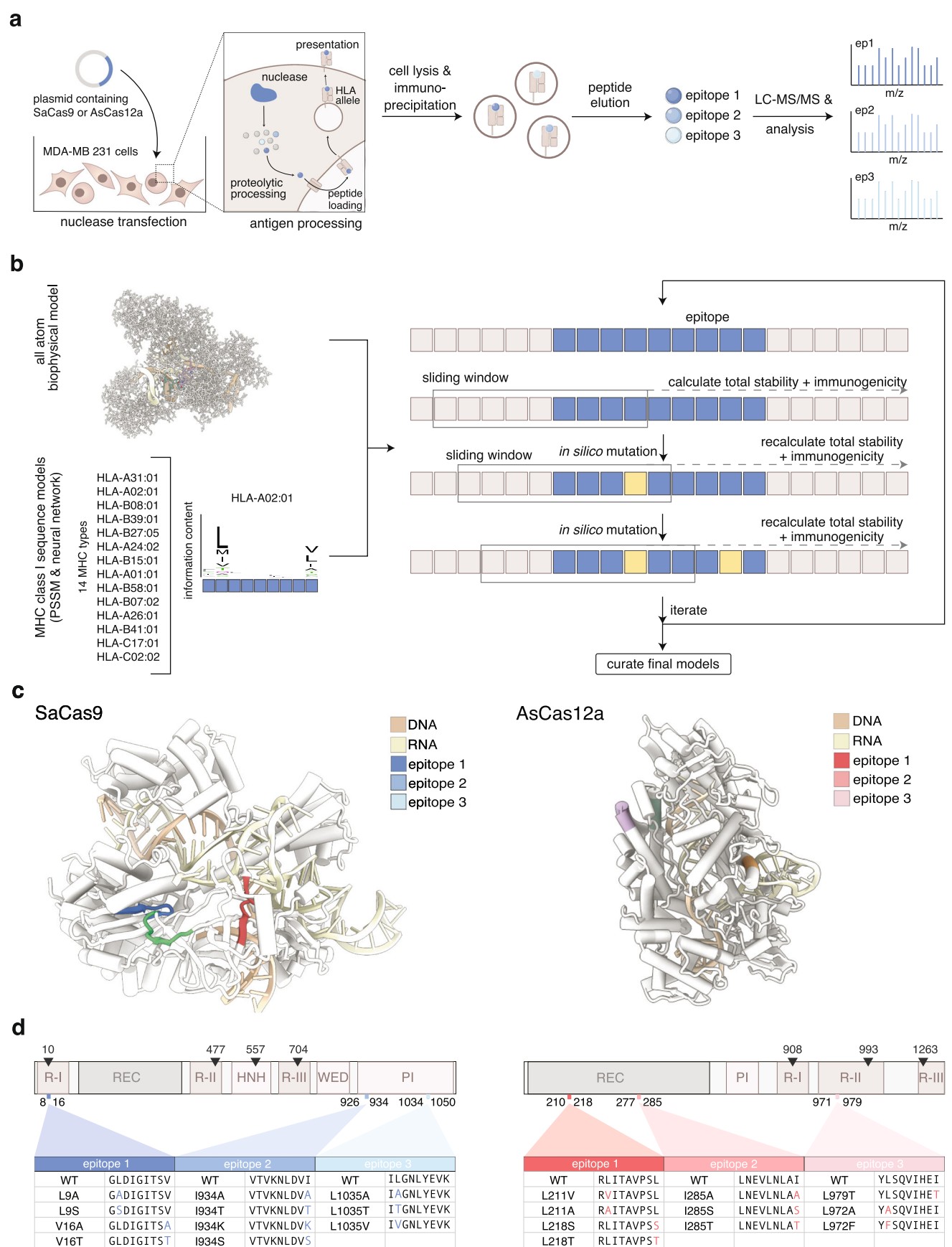

**Fig. 1 | Prediction of SaCas9 and AsCas12a epitopes with reduced binding to MHC I molecules. a** Schematic of MAPPs analysis to identify epitopes from SaCas9 and AsCas12a that bind to MHC I molecules. **b** Computational workflow to nominate mutations predicted to abrogate epitope binding to MHC I molecules while maintaining nuclease function. Crystal structures were used to create all-atom protein models in Rosetta. Epitope regions identified in MAPPs were targeted for mutational analysis, along with adjacent N-terminal and C-terminal subsequence frames to ensure that new epitopes were not created for any overlapping peptide subsequences. A computational protein design method utilized 14 MHC Class I PSSM models to introduce mutations predicted to eliminate MHC binding of epitope peptides while avoiding the creation of new predicted epitopes and maintaining predicted protein stability. Final models were evaluated using NetMHCpan and Rosetta. **c** Location of immunogenic epitopes on SaCas9 (left) and AsCas12a (right). **d** Sequences of immunogenic epitopes. Domain architecture of SaCas9 (left) and AsCas12a (right) with catalytic sites shown in red above and location of immunogenic epitopes indicated below. Sequences of immunogenic epitopes and proposed single amino acid mutations for each epitope are listed below R-I RuvC-I, REC recognition domain, R-II RuvC-II, HNH HNH nuclease, R-III RuvC-III, WED wedge domain, PI PAM-interacting domain.

## Engineered SaCas9 variants retain in vivo editing efficacy with minimal host immune response

We next investigated the efficiency and immunogenicity profiles of the SaCas9 Redi variants in vivo. To test adaptive immune responses against the SaCas9 Redi variants, we immunized humanized HLA-A*0201 HLA-DRA*0101 HLA-DRB1*0101-transgenic mice (A2.DR1), which lack mouse MHC and express human MHC class I and II complexes, with purified recombinant SaCas9 wild-type peptides to mimic Cas9 exposure[27]. We then assessed the adaptive immune response to either WT or SaCas9 Redi variant peptides by ELISpot, detection of intracellular cytokines in SaCas9-specific T cells and anti-SaCas9 IgG ELISA (Fig. 4a and Supplementary Fig. 7a). Across all biological replicates, T cell cytokine production upon peptide recall for 6 h against altered SaCas9 Redi epitopes was substantially reduced relative to WT SaCas9 epitopes, in both IFN-γ (Fig. 4b) and IL-2 measurements (Supplementary Fig. 7b), and all three epitopes contributed to the overall reduction of T cell responses against SaCas9 Redi variants (Fig. 4c). We validated these findings with ELISpot assays of ex vivo cultured splenocytes exposed to SaCas9 WT epitopes 1-3 and their respective SaCas9 Redi variants (Supplementary Fig. 8). In addition to T cellular responses, we found the development of SaCas9 WT epitope 1-specific antibodies in SaCas9-exposed mice (Fig. 4b), which is in line with reports of human B cell responses against this protein[16,21]. The IgG antibodies raised in these mice, however, consistently demonstrated decreased binding affinity against the Redi variant of epitope 1 (Fig. 4b).

To test if the observed attenuated adaptive immune responses against RediCas9 variants translate to improved efficacy in the context of repetitive dosing of a genome editing treatment, we administered AAV8 vectors, which target hepatocytes, encoding either WT SaCas9 or the SaCas9.Redi variants, along with a gRNA targeting *Pcsk9*, to A2.DR1 mice. The mice were re-treated after 14 days to simulate repetitive dosing. At day 21, we isolated splenocytes from the treated mice and stimulated them ex vivo with the respective immunogenic WT or variant epitopes for 24 h to evaluate the recall of an adaptive T cell response following exposure to these nucleases (Fig. 4d). For SaCas9 epitope 1, the L9A (SaCas9.Redi.1) and V16A mutations (SaCas9.Redi.3) resulted in a decreased T cell response in treated animals, whereas the L9S mutation (SaCas9.Redi.2) exhibited comparable immunogenicity to the WT epitope. For epitopes 2 and 3, all engineered mutations induced a less pronounced T cell response upon recall, mirroring our in vitro ELISpot findings in healthy human donors and animals exposed to purified SaCas9 protein variants (Fig. 4e). We found that repeated treatment with WT SaCas9 resulted in the generation of a T cell response in A2.DR1 mice, which was mitigated when using SaCas9.Redi.1, Redi.2, and Redi.3 variants (Fig. 4e and Supplementary Fig. 9a). At day 21, we additionally evaluated the editing efficiency in mouse liver (Fig. 4f). The indel rates at *Pcsk9* in the liver were comparable between WT SaCas9 and SaCas9.Redi.1 (both SaCas9.Redi.2 and SaCas9.Redi.3 showed lower editing than WT but still detectable editing levels) (Fig. 4f). LDL (low-density lipoprotein) cholesterol serum levels in mice treated with WT, SaCas9.Redi.1, and SaCas9.Redi.3 led to comparable reductions seen in previous studies in mice, nonhuman primates and early human data[28,29] (Fig. 4g, h).

SaCas9.Redi.2 conferred improved serum-LDL reduction compared to WT SaCas9 with repetitive dosing (Fig. 4g–i). Furthermore, no tested variants resulted in any short-term liver toxicity (Supplementary Fig. 9b). Lastly, we assessed the humoral immune response by multiplex cytokine detection in AAV8-WT-SaCas9 and AAV8-SaCas9.Redi-treated animals. Consistent with the observed reduced adaptive immunogenicity profile, treatment with AAV8-SaCas9.Redi.1 resulted in decreased serum levels of all tested pro-inflammatory cytokines compared to AAV8-WT-SaCas9, whereas treatment with both AAV8-SaCas9.Redi.1 and -2 resulted in decreased serum levels of specifically Interleukin-1β (Il-1β), which is typically produced by monocytes and macrophages in response to a various stimuli, including tissue damage[30–32]. Of note, all the AAV8-SaCas9 variants produced less GM-CSF and IL-2 elevation relative to the AAV-GFP control virus in this experiment (Supplementary Fig. 9c). Together, these data demonstrate that SaCas9.Redi.1 exhibited reduced immunogenicity without compromising its efficacy when compared to WT SaCas9 in vivo, highlighting its potential as a minimally immunogenic genome editing tool.

## Discussion

Recent studies have highlighted important considerations regarding the immune response to genome editing tools. For example, observational studies have reported a high prevalence of SpCas9 reactive T cells in the adult human population. Additionally, mice immunized against SaCas9 with Freund's adjuvant prior to intravenous delivery of AAV8-SaCas9-sgRNA exhibited CD8+ T cell accumulation in the liver and subsequent lysis of edited hepatocytes. Other mouse studies have similarly demonstrated that while the pre-existing SaCas9 antibodies did not prevent gene editing, there was a robust CD8+ T-cell response which eliminated the genome edited cells[22]. These findings underscore the potential influence of pre-existing immunity due to exposure to *S. aureus* under inflammatory conditions on the host response to Cas9 in humans[16,22,33]. Here, we establish a workflow to identify and remove immunogenic T cell epitopes while preserving wild-type levels of activity and specificity to address this challenge, generating a suite of reduced immunogenicity (Redi) variants of two clinically relevant nucleases, SaCas9 and AsCas12a.

To validate these Redi variants in vivo, we replicated the development of adaptive immunity to AAV8-SaCas9 by repetitive dosing of mice with AAV8-SaCas9-sgRNA at a 14-day interval. Consistent with previous studies, we observed substantial T cell immunity against SaCas9 WT epitopes, which was significantly reduced with the SaCas9.Redi.1 variant. Although it remains unclear how well immunized mouse models recapitulate pre-existing immunity in humans, we did observe a reduced CD8+ T cell response to this variant from human PBMCs. We demonstrated that regardless of HLA-B and HLA-C background, if an individual possesses the HLA-A*0201 allele, they have reduced CD8+ T cell reactivity to the engineered mutant epitopes. The HLA-A*0201 allele is one of the most common alleles and is present in 20–40% of Caucasians although it is less common in other populations. This points towards the broad applicability of these engineered Cas9 variants to reduce immunogenicity. We have

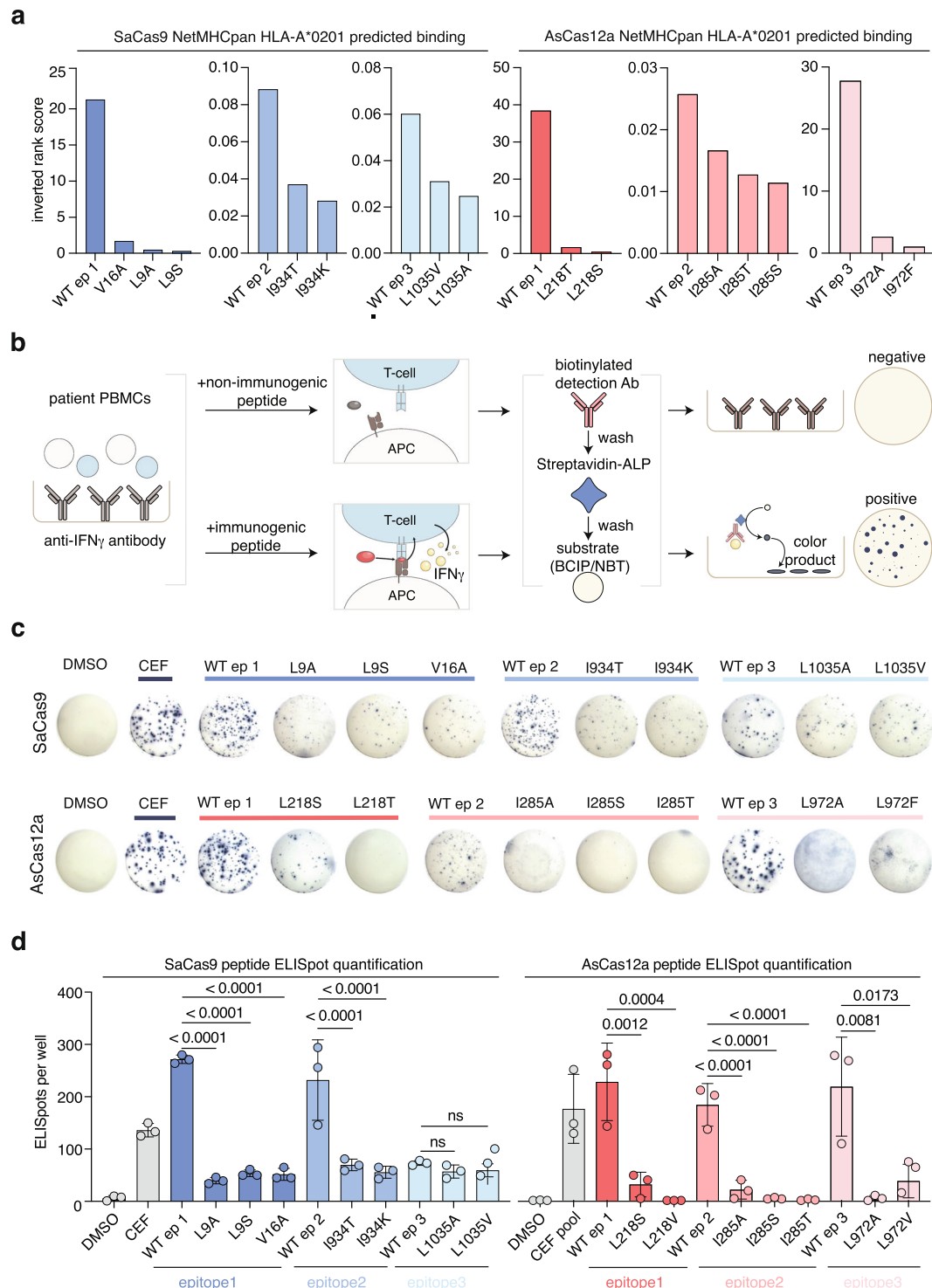

**Fig. 2 | SaCas9 and AsCas12a peptides with single point mutations are less immunogenic in vitro. a** Inverted rank scores for predicted binding between HLA-A*0201and SaCas9 (left) and AsCas12a (right) wild-type and predicted low-immunogenic peptides based on NetMHCpan 4.1 predictions. An inverted rank score >2 indicates strong binding and an inverted rank score <2 but >0.5 indicates weak binding. **b** Schematic of ELISpot assay. **c** Representative ELISpot images from peptide-treated PBMCs from HLA-A*0201 healthy donors (see Supplementary Fig. 3A, B for additional images). **d** Quantification of ELISpot images for SaCas9 (left) and AsCas12a (right). Plotted bars indicate mean ELISpot counts and error bars reflect the standard deviation across ELISpot spot counts for three technical replicates for each peptide condition. Significance comparisons were assessed using one-way ANOVA, and those comparisons that were significant at a $p$ value of 0.05 are shown with an asterisk (*), comparisons with a $p$ value < 0.001 are shown with two asterisks (**), and comparisons with a $p$ value < 0.0001 are shown with three asterisks (***). For SaCas9 epitope 1, $p$ values for comparisons of the mutant epitopes to WT ep1 (from left to right) are <0.0001 and <0.0001. For SaCas9 epitope 2, $p$ values for comparisons of the mutant epitopes to WT ep2 (from left to right) are <0.0001 and <0.0001. For SaCas9 epitope 3, $p$ values for comparisons of the mutant epitopes to WT ep3 (from left to right) are 0.1756 and 0.2508. For AsCas12a epitope 1, $p$ values for comparisons of the mutant epitopes to WT ep1 (from left to right) are 0.0012 and 0.0004. For AsCas12a epitope 2, $p$ values for comparisons of the mutant epitopes to WT ep2 (from left to right) are <0.0001 and <0.0001. For AsCas12a epitope 3, $p$ values for comparisons of the mutant epitopes to WT ep3 (from left to right) are 0.0081 and 0.0173. See also Source Data.

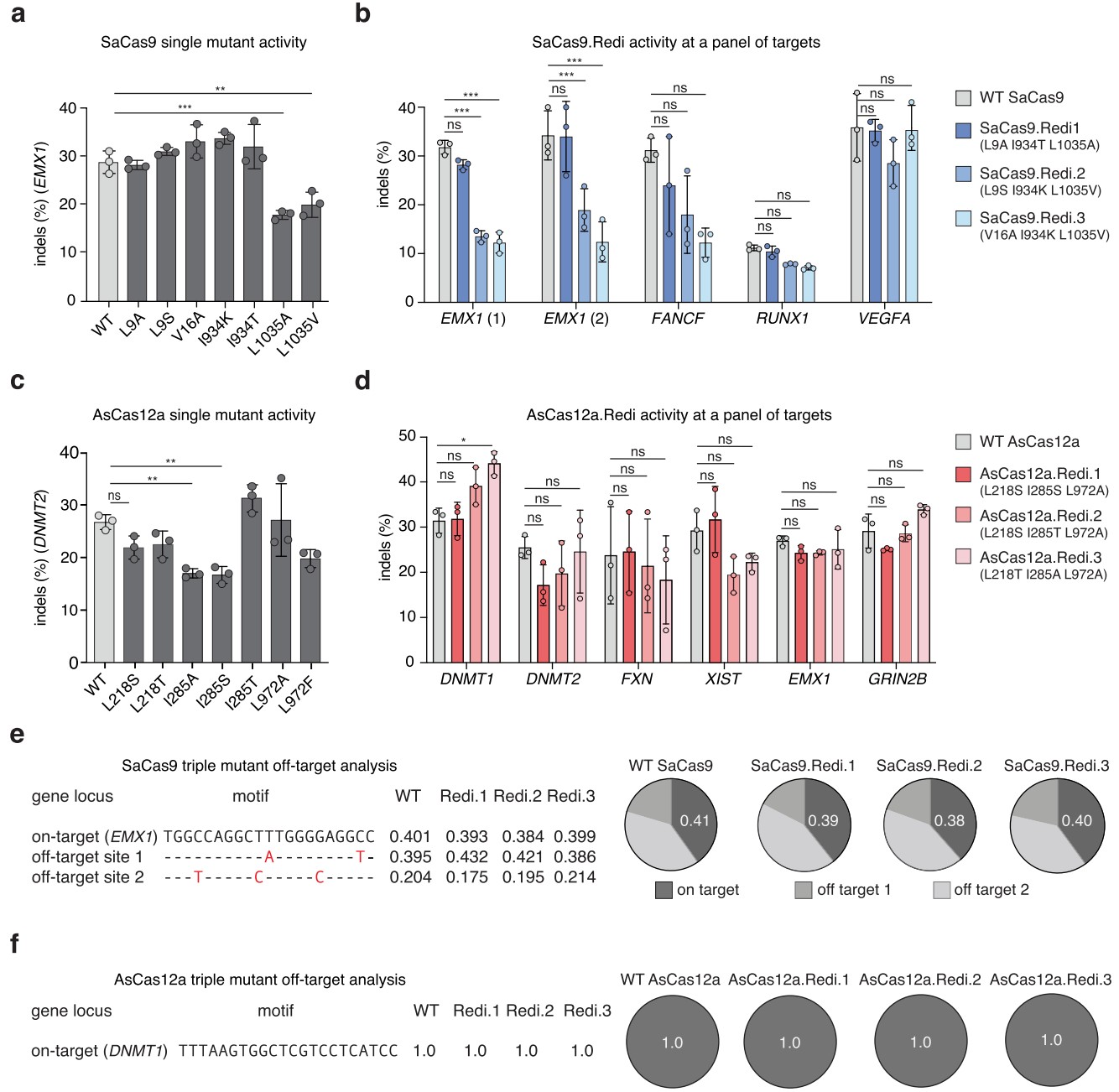

also shown computationally that we anticipate our Redi.Cas9 variants would have utility across a wide diversity of HLA-alleles (Supplementary Data 7, 8 and Supplementary Figs. 10 and 11) but to further enhance these variants, future work should focus on engineering nuclease variants that are minimally immunogenic across multiple HLA types, extending our work here with the HLA-A*0201 haplotype. It is also essential to thoroughly profile what proportion of gene therapy recipients will benefit from elimination of CD8+ T cell responses to these three epitopes. Development of these measures is essential for improving the safety and efficacy of genome editing therapies in diverse clinical settings, especially if multiple such treatments might be performed in a patient's lifetime and across different patient populations.

Additionally, in this work we focused primarily on abrogating cellular immunogenicity by reducing peptide presentation by MHC Class I but immune reactions to gene editing vectors encompass humoral immunogenicity as well. Our identified epitopes 2 and 3 map to immunodominant epitope sequences identified by MHC Class II

pulldown and antibody recognition reported previously[18]. Although our experimental data suggests some reduction in humoral immunity with our engineered variants, there are several other identified immunodominant epitopes that future work should focus on to achieve broad-spectrum reduction in both humoral and cellular immunogenicity while preserving functionality. Other approaches to avoid immune detection may include the exploration of Cas9 orthologs or conversion of Cas9 sequences to more human-like sequences. We hope that the structural and functional considerations and techniques used in this work will extend the clinical contexts in which Cas9 and Cas12 can be deployed as well as guide the iterative design of other therapeutic proteins with known structures.

## Methods

### Ethical statement

All experiments were performed in compliance with all relevant ethical regulations as approved by the Institutional Biosafety Committee (IBC) of the Broad Institute (Protocol # IBC-2017-00146) and the Office of

**Fig. 3 | SaCas9 and AsCas12a reduced immunogenicity (Redi) variants retain activity and specificity. a** Indel rates for wild-type (WT) SaCas9 and single-point mutant variants at *EMX1*. Plotted bars represent the mean indel rate and error bars represent standard deviation across three biological replicates. Significance comparisons were assessed using one-way ANOVA, and those comparisons that were significant at a *p* value of 0.05 are shown with an asterisk (*), comparisons with a *p* value < 0.001 are shown with two asterisks (**), and comparisons with a *p* value < 0.0001 are shown with three asterisks (***). For the *EMX1* target, *p* values (from left to right) were >0.9999, 0.9293, 0.3245, >0.9999, 0.1961, 0.6524, 0.003, and 0.003. See also Source Data. **b** Indel rates for WT SaCas9 and Redi variants at a panel of targets. Plotted bars represent the mean indel rate and error bars represent standard deviation across three biological replicates. SaCas9.Redi1 contains mutations L9A, I934T, L1035A. SaCas9.Redi.2. contains mutations L9S, I934K, and L1035V and SaCas9.Redi.3 contains mutations V16A, I934K, L1035V. Significance comparisons were assessed using one-way ANOVA, and those comparisons that were significant at a *p* value of 0.05 are shown with an asterisk (*), comparisons with a *p* value < 0.001 are shown with two asterisks (**), and comparisons with a *p* value < 0.0001 are shown with three asterisks (***). For the *EMX1* site 1 target, *p* values (from left to right) were 0.8062, <0.0001, <0.0001, and <0.0001. For the *EMX1* site 2 target, *p* values (from left to right) were >0.999, 0.0002, <0.0001, and <0.0001. For the *FANCF* target, *p* values (from left to right) were 0.1963, 0.0017, and <0.0001. For the *RUNX1* target, *p* values (from left to right) were 0.9994, 0.8456, and 0.7236. For the *VEGFA* target, *p* values (from left to right) were 0.9996, 0.1831, and <0.0001. See also Source Data. **c** Indel rates for WT AsCas12a and single-point mutant variants at *DNMT2*. Plotted bars represent the mean indel rate and error bars represent

standard deviation across three biological replicates. Significance comparisons were assessed using one-way ANOVA, and those comparisons that were significant at a *p* value of 0.05 are shown with an asterisk (*), comparisons with a *p* value < 0.001 are shown with two asterisks (**), and comparisons with a *p* value < 0.0001 are shown with three asterisks (***). For the *DNMT2* target, *p* values (from left to right) were 0.2851, 0.4052, 0.0068, 0.0053, 0.3256, >0.999, and 0.0647. See also Source Data. **d** Indel rates for WT AsCas12a and Redi variants at a panel of targets. Plotted bars represent the mean indel rate and error bars represent standard deviation across three biological replicates. Significance comparisons were assessed using one-way ANOVA, and those comparisons that were significant at a *p* value of 0.05 are shown with an asterisk (*), comparisons with a *p* value < 0.001 are shown with two asterisks (**), and comparisons with a *p* value < 0.0001 are shown with three asterisks (***). For the *DNMT1* target, *p* values (from left to right) were >0.999, 0.7926, and 0.2010. For the *DNMT2* target, *p* values (from left to right) were 0.4631, 0.8485, and >0.999. For the *FXN* target, *p* values (from left to right) were >0.999, 0.9986, and 0.8436. For the *XIST* target, *p* values (from left to right) were 0.9995, 0.4284, and 0.8101. For the *EMX1* target, *p* values (from left to right) were 0.9989, 0.9989, and >0.9999. For the *GRIN2b* target, *p* values (from left to right) were 0.9884, >0.999, and 0.9712. AsCas12a.Redi.1 contains mutations L218S, I285S, L972A. AsCas12.Redi.2 contains mutations L218S, I285T and L972A. AsCas12a.Redi.3 contains mutations L218T, I285A, and L972A. See also Source Data. TTISS off-target analysis for WT SaCas9 and Redi variants using an *EMX1*-targeting guide (**e**) and WT AsCas12 and Redi variants using a *DNMT1*-targeting guide (**f**). Numbers represent the fraction of reads with double-stranded DNA breaks that map to the given sequence. Note no off-targets were detected for Cas12. See also Source Data.

Research Subject Protection (ORSP) at the Broad Institute. Animal procedures followed the institutional laboratory animal research guidelines and were approved by the governmental authorities (Regional Administrative Authority Karlsruhe, Germany) or by the Institutional Animal Care and Use Committee (IACUC) of the Broad Institute (Protocol ID 0017-09-14-2).

### MHC-associated peptide proteomics (MAPPs)

MDA-MB-231 cells (#HTB-26, ATCC) were maintained in RPMI Medium supplemented with 10% Fetal bovine serum and 100 U/mL penicillin-streptomycin. Cells were expanded to 1.8e7 cells in a T225 flask and transfected with 60 μg of SaCas9 (#61591, Addgene) or AsCas12a (#69982, Addgene) using Lipofectamine 3000 (Thermo Fisher, L3000001) at 80% confluence. Media was changed 4 h post transfection to reduce toxicity and cells were harvested, washed 3x, and pelleted 48 h after transfection. Nuclease expression was confirmed by anti-HA (#3724S, Cell Signaling Technologies) western blot and surface MHC expression was confirmed by flow cytometry (#343330, Biolegend). MAPPs analysis was performed by Abzena.

### Isolation and analysis of HLA-ABC presented peptides from MDA-MB 231 cells

Ten million MDA-MB 231 cells were thawed at room temperature and subsequently lysed using a hypotonic buffer solution, 20 mM Tris, 5 mM MgCl 2 (Thermo Fisher Scientific), 0.1% Triton X-100 and protease inhibitors (Sigma-Aldrich), pH 7.8, for 1 h at 4 °C. HLA-A,B,C/peptide complexes were purified from the cell lysate by immunoprecipitation using magnetic beads (Promega, Southampton, UK) coated with anti-HLA-ABC antibody clone W6/32 (BioLegend, London, UK) overnight at 4 °C. Peptides bound to HLA-ABC were eluted under acidic conditions, 3% MeCN, 0.2% TFA in $H_2O$ (ThermoFisher Scientific) and purified by solid phase extraction using Oasis® HLB μElution plates (Waters, Ellsmere Port, UK). Peptides were freeze-dried using a 5301 vacuum concentrator (Eppendorf, Stevenage, UK) and stored at 80 °C until analyzed by MS. Freeze-dried peptides were re-solubilised in 3% MeCN, 0.2% TFA in $H_2O$ and analyzed using nano liquid chromatography coupled to an Exactive Plus™ mass spectrometer (all from ThermoFisher Scientific). Nano flow reverse phase

separation was performed using a Dionex Ultimate 3000 with an Acclaim™ PepMap™ 100 C18 separation column (75 μm × 150 mm, 2 μm, 100 Å) connected online to a Q Exactive Plus™ mass spectrometer (all from ThermoFisher Scientific) via a nano-spray ion source. Peptides were eluted at a flow rate of 300 nL/min using a linear gradient from 9% to 40% acetonitrile in 0.1% formic acid (both MS grade from ThermoFisher Scientific) over 105 min, followed by an increase from 40% to 65% acetonitrile in 0.1% formic acid over 20 min, with the column temperature set at 35 °C. The Q Exactive Plus™ mass spectrometer was operated in a top 10 data-dependent acquisition mode including lock mass (445.12003), for acquiring accurate mass (MS1 70,000; MS2 17,500 resolution) and HCD fragmentation, nCE 28, spectra (profile mode) of eluted peptides (AGC target 3e6, Maximum IT:100 ms, scan range 350 to 1100 m/z). Peptides were identified using the Sequest algorithm, built in the Proteome Discoverer software v2.1 (Thermo Fisher Scientific) against the reference human proteome TaxID: 9606v2017-04-12. Briefly, the search conditions used were: non-specific enzymatic cleavage, methionine oxidation and cysteinylation as variable modifications (maximum of one per peptide), 7.5 ppm parent ion and 0.025 Da fragment ion mass error. In alignment with literature[34] an FDR value of 1% was selected for validation of the peptide identified. Common contaminants were added to the database search and no filter was made on sequence length. Once the final list of identified peptides was completed, the sequence heatmaps were generated using MATLAB (MathWorks®, Cambridge, UK) to allow visualization of the sequence location and frequency of the identified peptides. For target protein only searches, data was searched using only the SaCas9 or AsCpf1 protein sequence and the spectra were manually inspected to verify the ID. The mass spectrometry proteomics data have been deposited to the ProteomeXchange Consortium via the PRIDE[35] partner repository with the dataset identifier PXD054579.

### Immunopeptidomics data quality analysis

Immunopeptidomic data quality analysis was performed using Immunolyser[36]. This analysis included peptide length distribution plots, upset plots depicting peptide overlap, GibbsCluster 2.0[37], and predicted number of binders to each HLA-allele. Immunolyser

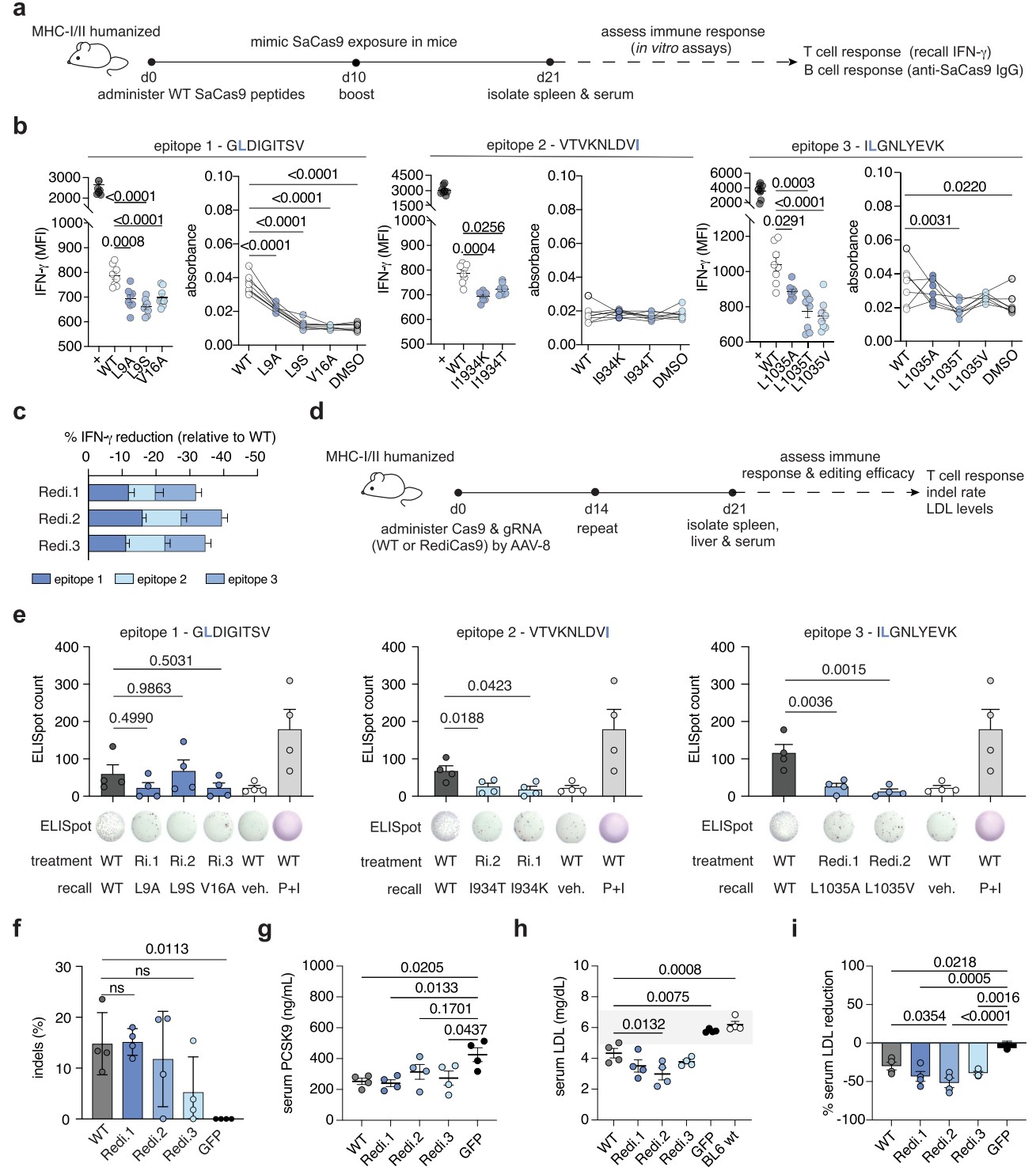

submits all selected HLA allotypes to MixMHCpred[38] and reports the best binding allotype for the peptide recommended by MixMHCpred. Contaminations and Decoy peptides were filtered out in pre-processing and peptides of length 5 to 30 residues were selected. For binding motif analysis using Gibbs clustering, logo plots were created by Seq2Logo[39] of the major peptide cluster. Peptide-HLA binding predictions were performed using the MHCflurry Class1PresentationPredictor in a Python environment. Peptide sequences were uploaded from a CSV file, and only 9-mer peptides were selected for analysis. The selected HLA alleles were HLA-A*0201, HLA-A*0217, HLA-B*4101, HLA-B*4002, HLA-C*1701, and HLA-C*0202. The predictor ran on filtered peptides, and binding was defined as an IC50 value ≤ 500 nM. Binding counts for each allele were calculated by grouping predictions by peptide and checking for affinity below the threshold, with non-binding peptides also recorded (code is provided as Supplementary Code).

**Computational mutation analysis**

Rosetta[40,41] was used to design and evaluate mutations in epitope regions identified in MAPPs. Crystal structures (PDB accessions 5AXW, 5B43) were downloaded from the PDB and refined in Rosetta. Mutations were scored with a modified Ref2015 score function[42] with additional score terms for evaluating peptide-MHC class I binding[43,44].

**Fig. 4 | SaCas9.Redi variants show reduced immunogenicity in vivo.**
**a** Experimental workflow to assess adaptive immunogenicity of WT SaCas9 or SaCas9.Redi variants in MHC-I/II-humanized (A2.DR1) mice. **b** T cell recall (left) and antibody titers (right) against WT epitopes 1–3 and their respective variants. T cell recall was performed by intracellular flow cytometry of IFNγ and IL-2 (Supplementary Fig. 7b) following overnight incubation of splenocytes from WT SaCas9-exposed MHC-I/II humanized mice with the indicated peptides. Anti-SaCas9 IgG levels were measured by ELISA. Absorbance (OD450 nm–570 nm) shown. $N = 8$ animals. Mean ± SEM shown. Statistical significance was determined by repeated-measures one-way ANOVA followed by Dunnett's post hoc test for multiple hypothesis correction. **c** Percent IFNγ reduction by each nuclease variant at d21 post-exposure. Relative proportions of mutated epitopes indicated by stacked bars. Mean ± SEM shown. $N = 8$ animals per group. **d** Workflow to assess editing efficiency of *Pcsk9* by WT SaCas9 or SaCas9.Redi variants. A2.DR1 mice were treated with 2e11 vg of AAV8-encoded nuclease variants at d0 and d14. Readouts were performed at d21 post-injection. **e** Quantification of SaCas9 T cell recalls split by epitope and respective variant mutation. IFN-γ ELISpot counts shown. Full counts in

Source Data. $N = 4$ animals per epitope. Mean ± SEM shown. Statistical significance was determined by one-way ANOVA followed by Sidak post hoc test for multiple hypothesis correction. See also Source Data. **f** *Pcsk9* indel rates measured in liver tissue at d21 post-injection. $N = 4$ biological replicates per condition. Mean with standard deviation is shown. Statistical significance was determined by one-way ANOVA followed by Dunnett test for multiple hypothesis correction. **g** Serum mouse PCSK9 measurements of treated animals at d21 post-injection by ELISA. Mean ± SEM shown. Statistical significance was determined by one-way ANOVA followed by Sidak post hoc test for multiple hypothesis correction. $N = 4$ animals per group. **h** Serum LDL cholesterol measurements of treated animals at d21 post-injection by ELISA. Mean ± SEM shown. Statistical significance was determined by one-way ANOVA followed by Sidak post hoc test for multiple hypothesis correction. $N = 4$ animals per group. **i** Percent serum LDL reduction at d21 post-injection by ELISA. Mean ± SEM shown. Statistical significance was determined by one-way ANOVA followed by Sidak post hoc test for multiple hypothesis correction. $N = 4$ animals per group.

---

PSSMs for a panel of 14 MHC class I alleles were created by calculating 9mer amino acid frequencies for all peptides in the IEDB[45] with a binding affinity less than 500 nM. These PSSMs were used for protein design using Rosetta Scripts. Final models were curated by manual inspection.

### NetMHCpan 4.1 predictions
NetMHCpan 4.1[46] was used to iteratively evaluate the effect of mutations on the binding affinity of epitope sequences to a panel of 14 MHC alleles (HLA-A*3101, HLA-A*0201, HLA-B*0801, HLA-B*3901, HLA-B*2705, HLA-A*2402, HLA-B*1501, HLA-A*0101, HLA-B*5801, HLA-A*2601, HLA-B*0702, HLA-B*4101, HLA-C*1701, HLA-C*0202). NetMHCpan 4.1 was used to predict binding affinities for all 9-mer peptides in the protein sequence in order to ensure that new predicted epitopes were not created by introduction of point mutations. The resulting predicted 9-mer peptides were run through NetMHCpan 4.1 to ensure that the new mutants would decrease predicted binding to all MHC alleles, including HLA-A*0201. Once new mutants were created, we expanded our NetMHC 4.1 predictions to 102 of the most enriched HLA alleles in the U.S population to evaluate for how many other HLA-alleles the new mutants would decrease predicted binding. The full list of profiled HLA-alleles can be found in Supplementary Fig. 10 as well as the corresponding frequencies of these alleles in the US population as determined from the Allele Frequency Database (https://www.allelefrequencies.net/). The data was normalized across each HLA-allele and for each epitope or mutant peptide, the sum of the frequencies for the HLA-alleles that yielded reductions in predicted peptide binding was computed and reported.

### Cloning and selection of single, double, and triple mutant nuclease variants
Single, double, and triple mutants were cloned for SaCas9 into construct pX601 (Addgene #61591) and for AsCas12a into construct pY010 (Addgene #69982). All mutations were cloned using site-directed mutagenesis using Phusion Flash for amplification and KLD for digestion and ligation. Guides were cloned into gRNA scaffold-containing constructs for SaCas9 (Addgene #70709) and AsCas12a (Addgene #pY020) respectively. Following assessment of indel activity for each single mutant, all mutants that retained activity that was 50% or greater than that of wild-type activity were selected as "top-performing" and double mutants were made with combinations of the single mutants. This was repeated for the creation and evaluation of the triple mutants. Specifically, for SaCas9, double mutants were made by creating the 'top-performing' mutations in epitope 1 and 2 and triple mutants were made by adding mutations in epitope 3 to the top-performing double mutants. For AsCas12a, double mutants were made as two-mutation combinations of all top-performing single mutations from unique

epitopes and triple mutants were created as three-mutation combinations at unique epitopes of the top-performing double mutations.

### HEK293FT cell line transfection
Human embryonic kidney 293FT cells (Life Technologies, R70007) were maintained in Dulbecco's modified Eagle's Medium (DMEM) supplemented with 10% FBS (HyClone), 2 mM GlutaMAX (Life Technologies), 100 U/ml penicillin, and 100 μg/ml streptomycin at 37 °C with 5% $CO_2$ incubation. Cells were seeded into 96-well plates (Corning) 1 day prior to transfection at a density of 150,000 cells per well and transfected at 70–80% confluency using Lipofectamine 3000 (Life Technologies) following the manufacturer's recommended protocol. For each well of a 96-well plate, a total of 200 ng DNA was used.

### DNA isolation from liver tissue
Genomic DNA was extracted using the QIAmp DNA mini kit (#51304, Qiagen) according to the manufacturer's instructions. Tissue samples were cut into smaller fragments from bulk tissue, resuspended in tissue lysis buffer with Proteinase K, and digested overnight at 56 °C. Following this, DNA was extracted using column-based purification and quantified on a NanoDrop.

### Sequencing and indel analysis
Following DNA extraction, 20 ng of gDNA per sample was used for the first PCR. The PCR products from the first reaction were then normalized and 20 ng of the purified product was annealed with the Illumina adapter and barcode sequences for the second PCR. The resulting product was isolated, purified, and analyzed using MiSeq. The primers used for PCR are in Supplementary Information. CRISPRESSOv2[47] was used to analyze the frequency of indels within each amplicon. Significance was assessed with a one-way ANOVA, and significant comparisons at a $p$ value of 0.05 are shown in the corresponding figure with an asterisk.

### In vitro ELISpot assay
We used HLA-A*0201 specific patient peripheral blood mononuclear cells (PBMCs) (LP_159, Cellular Technology Limited) to profile HLA-A2 patient specific CD8+ T cell reactivity for all ELISpot assays. Predicted SaCas9 and AsCas12a peptides, as listed in Fig. 2a, were synthesized from Genscript with >98% purity. Each peptide was reconstituted according to manufacturer's solubility instructions and made up to 1 mg/ml in $H_2O$. A human IFNγ precoated ELISpot kit was used to detect antigen-reactive T cells (EL285, R&D Systems). A total of 5e5 PBMCs were plated per well of a 96-well plate in ImmunoCult-XF T-cell Expansion media (#10981, STEMCELL Technologies) with recombinant IL-2 (#78036.3, STEMCELL technologies) and stimulated with 10 μg/ml of peptide in IFNγ precoated ELISpot plates for 48 h at 37 °C. Plates

were developed according to the manufacturer's instructions. DMSO was used as a negative control and HLA Class I Control peptide CEF pools at 10 µg/ml (#100-0675, STEMCELL Technologies) were used as a positive control.

## AAV production

For production of AAV virus, HEK293FT cells (Life Technologies) were maintained as described above in 150-mm plates. For each transfection, 10 µg of pAAV2/8 serotype packaging plasmid (#112864, Addgene), 12 µg of pAdDeltaF6 helper plasmid (#112867, Addgene), and 6 µg of ITR flanked plasmid carrying the nuclease construct of interest were added to 1 mL of serum-free DMEM. A total of 125 µL of PEI "Max" solution (1 mg/mL, pH = 7.1) was then added to the mixture and incubated at room temperature for 10 min. After incubation, the mixture was added to 25 mL of warm maintenance media and applied to each dish to replace the old growth media. Cells were harvested between 48 and 72 h post transfection by scraping and pelleting by centrifugation. The AAV2/8 (AAV2 ITR vectors pseudotyped with AAV8 capsid) viral particles were then purified from the pellet according to a previously published protocol[48]. Briefly, A solution containing 40% PEG 8000 (Promega) and 2.5 M NaCl was added to the medium at a 1:4 ratio. The mixture was incubated on ice for at least 2 h while gently rocked, followed by centrifugation at $3000 \times g$ for 30 min. The resulting large white pellet was resuspended in PBS and treated with DNase. Viral-containing fractions were identified using qPCR after an iodixanol gradient, then combined. Purification and concentration of the viral particles were performed using Zeba Spin Desalting Columns (Thermo Fisher Scientific).

## Tagmentation-based tag integration site sequencing

The target site specificity of SaCas9 and AsCas12a in the human genome was tested with Tagmentation-based tag integration site sequencing (TTISS)[26]. 2e5 HEK293FT cells in 12-well plates were co-transfected with combinations of SaCas9/AsCas12a expression plasmid (800 ng) and target gRNA expression plasmid (200 ng). After 3 days of incubation at 37 °C, the supernatant was removed and total DNA of the cells was extracted by DNeasy Blood & Tissue Kit (Qiagen 69506). The 2 µg of extracted DNA was tagmented by house-made loaded Tn5 in TAPS buffer (50 mM TAPS-NaOH pH8.5, 25 mM MgCl$_2$) in 62.5-µL reactions. Reactions were mixed with 312.5 µL PB buffer (Qiagen) and purified on a silica spin column. Tagmented DNA was eluted in 25 µL water, and amplified in 100 µL PCR reactions using KOD Hot Start Master Mix (Millipore 71842) under the following thermal cycling conditions: 1 cycle, 94 °C, 2 min; 12 cycles, 98 °C, 10 s, 60 °C, 30 s, 68 °C, 1 min; 1 cycle, 68 °C, 3 min; 4 °C hold. Three µL of this first PCR product was used as the template for each 50 µl-second PCR reaction: 1 cycle, 94 °C, 2 min; 20 cycles, 98 °C, 10 s, 65 °C, 30 s, 68 °C, 1 min; 1 cycle, 68 °C, 3 min; 4 °C hold (total 32 cycles for first and second PCR reactions). PCR products from 6 different experimental conditions were pooled together, purified, and 300–1000-bp fragments were enriched using a 2% agarose gel. After column purifications, the resulting libraries were sequenced using a NextSeq v2 kit (Illumina), 75 cycle kit with 59 forward cycles, 25 reverse cycles and Index1 8 cycles. Reads were mapped to human genome version hg38.2 using http://BrowserGenome.org[49] following the protocol outlined in Schmid-Burgk et al.[26]. The number of reads mapping to each site was used as a proxy to quantify the frequency of double-stranded DNA breaks outside the target region.

## SaCas9 immunization of MHC-I/II humanized mice

HLA-A*0201 HLA-DRA*0101 HLA-DRB1*0101 transgenic mice devoid of mouse MHC [A2.DR1 mice, B6-Tg(HLA-DRA*0101, HLA-DRB1*0101)$^{1Dmz}$ Tg(HLA-A/H2-D/B2M)$^{1Bpe}$ H2-Ab1$^{tm1Doi}$ B2m$^{tm1Unc}$ H2-D1$^{tm1Bpe}$] were provided by M. Berard and bred at the DKFZ and Broad Institute animal facilities[27]. Female and male littermate mice were housed under

Specific and Opportunistic Pathogen Free (SOPF) conditions and 12-h day/night cycles. All experiments were conducted with sex-matched animals, without bias to either sex. Sex-based analysis was not performed. AAV SaCas9 delivery experiments were approved by the governmental authorities (Regional Administrative Authority Karlsruhe, Germany) overseeing the German Cancer Research Center (DKFZ). SaCas9 peptide vaccination experiments were approved by the Institutional Animal Care and Use Committee (IACUC) of the Broad Institute (Protocol ID 0017-09-14-2). Animal maintenance complied with all relevant ethical regulations and were consistent with local, state and federal regulations as applicable, including the National Institutes of Health Guide for the Care and Use of Laboratory Animals. A2.DR1 mice were immunized with 100 µg SaCas9 wild-type peptides emulsified in complete Freund's adjuvant (CFA, BD Biosciences). CFA was emulsified with equal volume of peptide in PBS to 1 mg ml$^{-1}$ or DMSO in PBS (sham) and mice received 2 subcutaneous injections of 50 µl each into the lateral pectoral regions. Mice were boosted after 10 days with 1 subcutaneous injection of 50 µl. No rmGM-CSF was applied at boost. Vaccination only experiments were terminated after 21 days.

## In vivo AAV treatment and processing

HLA-A*0201 HLA-DRA*0101 HLA-DRB1*0101 transgenic mice devoid of mouse MHC [A2.DR1 mice, B6-Tg(HLA-DRA*0101, HLA-DRB1*0101)$^{1Dmz}$ Tg(HLA-A/H2-D/B2M)$^{1Bpe}$ H2-Ab1$^{tm1Doi}$ B2m$^{tm1Unc}$ H2-D1$^{tm1Bpe}$] were provided by M. Berard and bred at the DKFZ animal facility[27]. Mice were housed under Specific and Opportunistic Pathogen Free (SOPF) conditions and 12-h day/night cycles. All animal procedures followed the institutional laboratory animal research guidelines and were approved by the governmental authorities (Regional Administrative Authority Karlsruhe, Germany). AAV vectors were delivered to 12–14-week-old male A2.DR1 mice intravenously via lateral tail vein injection. An absolute dose of 2e11 vg of the respective AAV vector per treatment was administered to each animal. Fourteen days after the first treatment, mice were re-treated with 2e11 vg of the respective AAV vector. All dosages of AAV were adjusted to 100 µL volume with sterile phosphate buffered saline (PBS), pH 7.4 (Gibco) before the injection. Mice received 50 µg agonistic αCD40 antibody (FGK4.5, Bioxcell) and 50,000 iU hIL2 (Roche) at the first and second immunization as well as 24 h after the first immunization. Animals were not immunosuppressed or otherwise handled differently prior to injection or during the course of the experiment. The animals were randomized to the different experimental conditions, with the investigator not blinded to the assignments.

For terminal procedures to organs and larger serum volumes for chemistry panels, mice were euthanized by terminal Ketamine/Xylazine injection. Subsequently, blood was collected via cardiac puncture, followed by collection of the spleen. Transcardial perfusion with 30 mL PBS removed the remaining blood, after which liver and lymph node samples were collected. The median lobe of liver was removed and snap-frozen on dry-ice for subsequent DNA extraction.

## T cell recall from SaCas9 protein-exposed or AAV-treated mice by ELISpot

Spleens were mashed through a 70-µm strainer. Contaminating erythrocytes were lysed using ACK lysis buffer (Gibco). ELISpot was performed as previously described[50]. Briefly, wetted ELISpot plates (MAIPSWU10, Millipore) were incubated with 100 µL of 15 µg/mL IFNγ coating antibody (AN-18, Mabtech) and incubated overnight at 4 °C. Cells were resuspended in RPMI-1640 supplemented with 10% FBS, P/S as above, 50 µmol/L beta-mercaptoethanol (Sigma), 2 mmol/L L-glutamine, 25 mmol/L Hepes, 1 mmol/L sodium pyruvate (all Invitrogen), and 0.1 mmol/L nonessential amino acids (NEAA, Lonza; called T-cell medium, TCM). IFNγ coating antibody was removed, and ELISpot plates were blocked with TCM. 500,000 cells were plated, and

peptides were added at 1 µg/mL. As a positive control, 20 ng/mL phorbol 12-myristate 13-acetate (PMA) and 1 µg/mL ionomycin or anti-CD3/CD28 stimulation beads (Gibco) were used. Plates were incubated for 72 h. Cells were removed and plates were incubated with 1 µg/mL biotinylated IFNγ detection antibody (R4–6A2, Mabtech) in PBS with 0.5% FBS for 2 h at room temperature. The detection antibody was removed, and wells were incubated with 1 µg/mL streptavidin–alkaline phosphatase (ALP; Mabtech) in PBS with 0.5% FBS for 1 h. Streptavidin-ALP was removed, and the plate was incubated with ALP development buffer (Bio-Rad) until distinct spots emerged. Spots were quantified with an ImmunoSpot Analyzer (Cellular Technology Ltd).

### T cell recall from SaCas9 protein-exposed or AAV-treated mice by intracellular flow cytometry

Splenocytes were prepared as described above and resuspended in RPMI-1640 supplemented with 10% FBS, P/S as above, 50 µmol/L beta-mercaptoethanol (Sigma), 2 mmol/L L-glutamine, 25 mmol/L Hepes, 1 mmol/L sodium pyruvate (all Invitrogen), and 0.1 mmol/L non-essential amino acids (NEAA, Lonza; called T-cell medium, TCM). A total of 500,000 cells per condition were incubated for 6 h at 37 °C in the presence of 1 µg/mL wild-type SaCas9 epitopes and respective engineered altered peptides. For the first 4 h of the 6-h incubation, GolgiPlugTM (51-2301KZ, BD Biosciences) was used to inhibit cytokine secretion. After the incubation, cells were washed twice with PBS, Fc receptors were blocked with anti-mouse-CD16/CD32 antibody (clone 93, BioLegend), and cells were stained with conjugated flow cytometry antibodies CD45 (BV510, clone 30-F11, BioLegend), CD3 (FITC, clone 17A2, BioLegend), CD8a (eFluor450, clone 53–6.7, Invitrogen), and CD4 (PerCP, clone RM4-5, BD Biosciences) for 20 min at 4 °C. A fixable viability dye (eFluor 780, 65-0865-14, Invitrogen) was used. After fixation with the BD fixation/permeabilization solution kit (554715, BD Biosciences) according to the manufacturer's instructions, intracellular cytokine staining was performed using anti-mouse-IFNγ (APC, clone XMG1.2, BioLegend) and IL-2 (PE, clone JES6-5H4, BioLegend) antibodies for 45 min at 4 °C. Following two more washes with PBS, cells were resuspended in PBS + 1 mM EDTA + 0.5% BSA and analyzed directly on a Beckman Coulter CytoFlex LX flow cytometer.

### Detection of antibodies against SaCas9 epitopes

For the detection of SaCas9 epitope-specific IgG, 10 µg/mL wild-type SaCas9 epitope 1, 10 µg/mL wild-type SaCas9 epitope 2 and 10 µg/mL wild-type SaCas9 epitope 3 in ELISA coating buffer (9052, Chondrex Inc) were immobilized on highly absorbent ELISA plates by overnight incubation at 4 °C. ELISA plates were washed twice with PBS + 0.1% Tween-20 (wash buffer), followed by an incubation with ELISA blocking buffer (30105, Chondrex Inc) for 1 h at room temperature. Plates were washed twice with a wash buffer and mouse serum samples were diluted 1:1000 and incubated for 1 more hour at room temperature. Detection was performed using peroxidase-conjugated polyclonal anti-mouse-IgG antibodies (30113, Chondrex Inc) at a 1:200 dilution followed by incubation for 1 more hour at room temperature and, lastly, chromogenic development based on the manufacturer's instructions. Absorbance data were collected by OD450nm-570nm measurement.

### Multiplex cytokine detection from mouse sera

Heart blood was aspirated from sacrificed mice using EDTA-precoated syringes and serum was isolated by centrifugation at 1000–2000 × g for 10 min in a refrigerated centrifuge. Serum samples were then diluted 1:200 with PBS and tested using the Milliplex MAP Mouse Cytokine/Chemokine Magnetic Bead Panel (Millipore, Cat No. MCY-TOMAG-70K-PMX) according to the manufacturer's protocol. Data was collected on a Luminex detection system (Luminex) and data analysis was completed using the BELYSA® 1.1.0 software. The data collected by

the instrument software are expressed as Median Fluorescence Intensity (MFI). MFI values for each analyte were collected per each individual sample well. Analyte standards, quality controls, and sample MFI values were adjusted for background. Calibrator data were fit to either a five-parameter logistic (5PL) or four parameter logistic (4PL) model depending on best fit to produce accurate standard curves for each analyte. Quality control and sample data were interpolated from the standard curves and then adjusted according to dilution factor to provide calculated final concentrations of each analyte present in the sample.

### LDL serum ELISA

Serum samples were diluted 1:100 with PBS and tested using the Mouse LDL ELISA Kit (Colorimetric) (Novus Biologicals, Cat No. NBP2-81135) according to the manufacturer's protocol. Data were collected by OD450 nm–570 nm measurement. LDL concentrations were calculated corresponding to diluted standards and control samples and were fit to a linear model to interpolate serum concentrations.

### Statistics and reproducibility

For ELISpot in vitro experiments, each condition had three biological replicates. No statistical method was used to predetermine sample size. Wells that had too many spots to count were excluded from analysis as the individual spots were not clearly resolved, precluding an accurate reading. The investigators were not blinded to allocation during experiments and outcome assessment. Statistical analysis was performed via a one-way ANOVA with a threshold $p$ value of 0.05. For indel sequencing experiments, each condition had three biological replicates and was reproducibly shown across three separate experiments to demonstrate reproducibility. No statistical method was used to predetermine sample size. No data were excluded from the analysis. The investigators were not blinded to allocation during experiments and outcome assessment. Statistical analysis was performed via a one-way ANOVA with a threshold $p$ value of 0.05. For TTISS experiments, each sample was run in duplicate for a single, well-characterized target. No statistical method was used to predetermine sample size. No data were excluded from the analysis. The investigators were not blinded to allocation during experiments and outcome. For animal experiments, the sample size was decided based on prior publications with similar experiments by our laboratory or others as indicated in the methods section of the manuscript. No statistical method was used to predetermine sample size. No animals or data were excluded from the analyses. For all experiments, allocation of mice into experimental groups was randomized after matching for age and sex. Separate investigators performed treatment and data collection. Data collecting investigators were blinded to the treatment groups. Data analyzing investigators were not blinded to the treatment groups, as they involved internal controls. Experimental and control animals were treated equally.

### Reporting summary

Further information on research design is available in the Nature Portfolio Reporting Summary linked to this article.

## Data availability

All data supporting the findings of this study are available within the paper and its Supplementary Information. The mass spectrometry proteomics data have been deposited to the ProteomeXchange Consortium via the PRIDE partner repository with the dataset identifier PXD054579 . Source data are provided with this paper.

## Code availability

Python script for immunopeptidomics analysis is provided in Supplementary Code.

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

## Acknowledgements

We thank A. Trocha and B. Walker at the Ragon Institute for assistance with imaging and quantification of ELISpot plates, J. Abelin for help with the immunopeptidomics data analysis, and all members of the Zhang lab for helpful discussions and support. This study was supported by the Federal Ministry of Education and Research (BMBF) and the Ministry of Science Baden-Württemberg within the framework of the Excellence Strategy of the Federal and State Governments of Germany (Project-ID ExU 6.1.12 to M.J.F.) and Deutsche José Carreras Leukämie-Stiftung (Project-ID DJCLS 01ZI/2022 to M.J.F.). R.R. is supported by the National Science Foundation's (NSF) Graduate Research Fellowship Program (GRFP). M.J.F. is a member of the MD/PhD program at Heidelberg University and a fellow of the German Research Foundation (DFG, grant ID FR 4701/1-1). F.Z. is supported by the Howard Hughes Medical Institute; the Poitras Center for Psychiatric Disorders Research at MIT; the Hock E. Tan and K. Lisa Yang Center for Autism Research at MIT; and the K. Lisa Yang and Hock E. Tan Molecular Therapeutics Center at MIT.

## Author contributions

I.K., L.N., and F.Z. conceived the study. R.R., M.J.F., I.K., and F.Z. designed the experiments. R.R., M.J.F., I.K., S.C.-D.-T.V., D.S., B.L., M.K., and M.P. performed experiments and analyzed data. F.Z., R.K.M., Y.S., and L.N. supervised the work. R.R., M.J.F., R.K.M., and F.Z. wrote the manuscript with input from all authors.

## Competing interests

F.Z., R.R., R.K.M., Y.S., L.N., and I.K. are co-inventors on a patent ("ENGINEERED TYPE II CAS POLYNUCLEOTIDES WITH REDUCED IMMUNOGENICITY AND USES THEREOF" (PCT/US2024/022898)) related to this work filed by the Broad Institute, MIT, and Cyrus Biotechnology. L.N. and Y.S. are executives at and co-founders of, and I.K. a senior scientist at, Cyrus Biotechnology. F.Z. is a scientific advisor and cofounder of Editas Medicine, Beam Therapeutics, Pairwise Plants, Arbor Biotechnologies, Aera Therapeutics, and Moonwalk Biosciences. F.Z. is a scientific advisor for Octant. M.J.F. reports speaker honoraria from Pfizer, Roche, and Kerna Ventures. The remaining authors declare no competing interests.
