## [Transparent Peer Review file · Nature Communications]

Rational engineering of minimally immunogenic nucleases for gene therapy

Corresponding Author: Professor Feng Zhang

Version 0:

Reviewer comments:

Reviewer #1

(Remarks to the Author)

The work by Raghavan et al seeks to generate minimally immunogenic nucleases based on SaCas9 and AsCas12a for human gene therapy. It is widely recognized that there is a high prevalence of both humoral and T-cell immunity to most Cas9 orthologs in humans, which could adversely affect the efficacy and safety of human gene therapies. This is especially true in cases where there is persistent expression of the nuclease, as with viral vectors. In this paper, the authors perform MAPPs analysis to identify immunogenic peptides in SaCas9 and AsCas12a. They then engineer variants of these nucleases with peptides that limit binding and presentation by human MHC. Through protein engineering, the authors generate SaCas9-Redi (reduced immunogenicity) which has less reactivity to three of the peptides identified relative to wild type SaCas9. Editing activity was preserved with the Redi variants, with no detectable increase in off-target activity. There are still some concerns about the broad applicability of such an approach given human HLA diversity, as well as individual responses to full length Cas9 exposure that generate pre-existing immunity. Nonetheless, this paper is an impressive effort which demonstrates that multiple immunogenic epitopes can be successfully modified whilst still preserving editing activity in vitro and in vivo.

Comments

-Figure 4 appears first in the PDF file, and is out of order.

-Reviewers' previous concerns were effectively addressed.

Reviewer #2

(Remarks to the Author)

This is a re-review of a minimally edited version of the same manuscript. This reviewer appreciates the effort of the team in providing a comprehensive in silico evaluation of antigenic peptide presentability, and is more convinced than before on the engineering approach and real-world applicability of this work.

Considering the broad audience of Nature Communications, this is now appropriate for publication, and I envision this article will be well-cited for the far-reaching conceptual advance that can be cross-applied to the engineering of other therapeutic protein variants.

Reviewer #3

(Remarks to the Author)

The manuscript by Raghavan and colleagues seeks to reduce the immunogenicity of bacterially derived Cas-9 nucleases by mapping potential MHC binding peptides derived from these enzymes and mutating key anchor residues to remove ensuing immune responses when these enzymes are used therapeutically. The authors in the revised version of the manuscript have computationally probed the potential reactivity of the engineered nucleases against several other HLA allotypes. This is

a step in the right direction but I feel more needs to be done to highlight the limitations of the approach. I do not wish to detract from the protein engineering aspect of the study and the conceptual advances made in this study and also feel that Nat Comms is a better destination for this calibre of work.

However, the failure to (still) appropriately report the immunopeptidomics analysis is problematic – whilst raw data is deposited in PRIDE there is little reporting of data quality and the standard metrics that go alongside such analysis (beyond a FDR filter) such as peptide length distribution and binding motif analysis. It is also not clear how well NETMHC predictions coincided with the isolated peptides and the quality with which HLA-A2 binders were determined since a pan class I monoclonal antibody was used to isolate peptides from polyallelic MBA-MD-231 cells. Tools such as Gibbscluster 2.0 could be used to cluster peptides to give an overall feel for the depth of immunopeptidomics coverage of the HLA A2 peptides. Fig 1a does not really portray the data generated and since this is the only evidence of natural presentation significantly more attention needs to be paid to this information before publication can be considered.

Version 1:

Reviewer comments:

Reviewer #3

(Remarks to the Author)

This is the third time I have reviewed this manuscript and am pleased to see considerable improvement in how the immunopeptidomics component of the analysis has improved. The level of reporting is now sufficient to allow better insight into the data quality. I now believe the manuscript should be accepted for publication without further delay.

(Remarks on code availability)

I am not python user and suggest a bioinformatician review the code if required.

REVIEWER COMMENTS

Reviewer #1 (Remarks to the Author):

The work by Raghavan et al seeks to generate minimally immunogenic nucleases based on SaCas9 and AsCas12a for human gene therapy. It is widely recognized that there is a high prevalence of both humoral and T-cell immunity to most Cas9 orthologs in humans, which could adversely affect the efficacy and safety of human gene therapies. This is especially true in cases where there is persistent expression of the nuclease, as with viral vectors. In this paper, the authors perform MAPPs analysis to identify immunogenic peptides in SaCas9 and AsCas12a. They then engineer variants of these nucleases with peptides that limit binding and presentation by human MHC. Through protein engineering, the authors generate SaCas9-Redi (reduced immunogenicity) which has less reactivity to three of the peptides identified relative to wild type SaCas9. Editing activity was preserved with the Redi variants, with no detectable increase in off-target activity. There are still some concerns about the broad applicability of such an approach given human HLA diversity, as well as individual responses to full length Cas9 exposure that generate pre-existing immunity. Nonetheless, this paper is an impressive effort which demonstrates that multiple immunogenic epitopes can be successfully modified whilst still preserving editing activity in vitro and in vivo.

Comments

-Figure 4 appears first in the PDF file, and is out of order.

-Reviewers' previous concerns were effectively addressed.

| We thank the reviewer for this positive feedback.

Reviewer #2 (Remarks to the Author):

This is a re-review of a minimally edited version of the same manuscript. This reviewer appreciates the effort of the team in providing a comprehensive in silico evaluation of antigenic peptide presentability, and is more convinced than before on the engineering approach and real-world applicability of this work.

Considering the broad audience of Nature Communications, this is now appropriate for publication, and I envision this article will be well-cited for the far-reaching conceptual advance that can be cross-applied to the engineering of other therapeutic protein variants.

| We thank the reviewer for this positive feedback.

Reviewer #3 (Remarks to the Author):

The manuscript by Raghavan and colleagues seeks to reduce the immunogenicity of bacterially derived Cas-9 nucleases by mapping potential MHC binding peptides derived from these

enzymes and mutating key anchor residues to remove ensuing immune responses when these enzymes are used therapeutically. The authors in the revised version of the manuscript have computationally probed the potential reactogenicity of the engineered nucleases against several other HLA allotypes. This is a step in the right direction but I feel more needs to be done to highlight the limitations of the approach. I do not wish to detract from the protein engineering aspect of the study and the conceptual advances made in this study and also feel that Nat Comms is a better destination for this calibre of work.

However, the failure to (still) appropriately report the immunopeptidomics analysis is problematic – whilst raw data is deposited in PRIDE there is little reporting of data quality and the standard metrics that go alongside such analysis (beyond a FDR filter) such as peptide length distribution and binding motif analysis. It is also not clear how well NETMHC predictions coincided with the isolated peptides and the quality with which HLA-A2 binders were determined since a pan class I monoclonal antibody was used to isolate peptides from polyallelic MDA-MB-231 cells. Tools such as Gibbscluster 2.0 could be used to cluster peptides to give an overall feel for the depth of immunopeptidomics coverage of the HLA A2 peptides. Fig 1a does not really portray the data generated and since this is the only evidence of natural presentation significantly more attention needs to be paid to this information before publication can be considered.

We thank the reviewer for this thoughtful input. We have added Extended Data Figures 1 and 2 to report on the data quality of immunopeptidomics. In this analysis, we include peptide length distribution and binding motif analysis. We performed the binding motif analysis with Gibbs Cluster 2.0 as recommended by the reviewer to provide more information about the peptide list. Additionally, to answer the question of the quality with which HLA-A2 binders were determined, we ran the 9-mers in the peptide list through MHCFlurry and found how many of the peptides were predicted to bind to HLA-A0201 (and the other 5 alleles in MDA-MB-231 cells). The python script to produce these findings has been provided for reference. We report the number of predicted binders in the total peptide set to each of the HLA alleles in MDA-MB-231 cells as well as the number of binders not predicted to bind to any of the 6 alleles. Through this analysis we see that a large majority of the 9-mers in the peptide list are predicted to be HLA-A0201 restricted even with the use of a pan class I monoclonal antibody with a polyallelic cell line. This further motivates the use of HLA-A0201 as the major focus of the protein engineering efforts that follow in the rest of the paper. We have amended Figure 1a to more accurately reflect the MAPPS analysis process.